# Epidemiological Characteristics of Human and Animal Plague in Yunnan Province, China, 1950 to 2020

Haonan Han,[a] Yun Liang,[b] Zhizhong Song,[c] Zhaokai He,[a,d] Ran Duan,[a] Yuhuang Chen,[e] Zihou Gao,[b] Shuai Qin,[a] Junrong Liang,[a] Deming Tang,[a] Dongyue lv,[a] Peng Zhang,[a] Dan Zhang,[a] Huaiqi Jing,[a] Xin Wang[a]

aState Key Laboratory of Infectious Disease Prevention and Control, National Institute for Communicable Disease Control and Prevention, Chinese Center for Disease Control and Prevention, Changping, Beijing, People's Republic of China
bYunnan Institute of Endemic Diseases Control and Prevention, Dali, Yunnan, People's Republic of China
cYunnan Provincial Center for Disease Control and Prevention, Kunming, Yunnan, People's Republic of China
dInstitute of Infectious Disease Control and Prevention, Hangzhou Municipal Center for Disease Control and Prevention, Hangzhou, Zhejiang, People's Republic of China
eShenzhen Nanshan Maternity and Child Healthcare Hospital, Shenzhen, Guangdong, People's Republic of China

Haonan Han, Yun Liang, Zhizhong Song, Zhaokai He, Ran Duan, Yuhuang Chen, and Zihou Gao contributed equally to this work. Author order was determined on the basis of seniority.

**ABSTRACT** This study analyzed the epidemiological characteristics of 3,464 human plague cases and the distribution pattern of 4,968 *Yersinia pestis* isolates from humans, hosts, and vector insects from 1950 to 2020 among two natural plague foci in Yunnan Province, China. These foci include the *Rattus flavipectus* plague focus of the Yunnan, Guangdong, and Fujian provinces and the *Apodemus chevrieri-Eothenomys miletus* plague focus of the highlands of northwestern Yunnan Province. The case fatality rate for plague in humans was 18.39% (637/3,464), and the total isolation rate of *Y. pestis* was 0.17% (4,968/2,975,288). Despite that the frequency of human cases declined rapidly, the animal plague fluctuated greatly, alternating between activity and inactivity in these foci. The tendency among human cases can be divided into 4 stages, 1950 to 1955, 1956 to 1989, 1990 to 2005, and 2006 to 2020. Bubonic plague accounted for the majority of cases in Yunnan, where pneumonic and septicemic plague rarely occurred. The natural plague foci have been in a relatively active state due to the stability of local ecology. Dense human population and frequent contact with host animals contribute to the high risk of human infection. This study systematically analyzed the epidemic pattern of human plague and the distribution characteristics of *Y. pestis* in the natural plague foci in Yunnan, providing a scientific basis for further development and adjustment of plague prevention and control strategies.

**IMPORTANCE** Yunnan is the origin of the third plague pandemic. The analysis of human and animal plague characteristics of plague foci in Yunnan enlightens the prevention and control of the next plague pandemics. The plague characteristics of Yunnan show that human plague occurred when animal plague reached a certain scale, and strengthened surveillance of animal plague and reducing the density of host animals and transmission vectors contribute to the prevention and control of human plague outbreaks. The phenomenon of alternation between the resting period and active period of plague foci in Yunnan further proves the endogenous preservation mechanism of plague.

**KEYWORDS** epidemiological characteristics, *Yersinia pestis* isolates, human plague cases, China, Yunnan

Address correspondence to Xin Wang, wangxin@icdc.cn.

The authors declare no conflict of interest.

Throughout history, plague pandemics have occurred many times around the world, causing hundreds of millions of deaths (1–10). Plague outbreaks also caused significant problems in China. In the first half of the 20th century, plague epidemics occurred

in China, and the death toll reached nearly 1 million (11–17). Based on the characteristics of the host, vector, geography, and *Y. pestis* ecotype, the natural plague foci in China can be divided into 12 types (18). Of these, two natural plague foci have been identified and classified in Yunnan Province, the *Rattus flavipectus* plague focus of the Yunnan, Guangdong, and Fujian provinces and the *Apodemus chevrieri-Eothenomys miletus* plague focus of the highlands of northwestern Yunnan Province, and these two foci cover approximately 79,216 km². Since the 1990s, there has been a large-scale human plague epidemic in the *Rattus flavipectus* plague focus of the Yunnan, Guangdong, and Fujian provinces, and new cases arise every few years (19). Due to limitations in experimental technology and economic conditions, most plague diagnoses that occurred in China prior to the 1950s were dependent on clinical symptoms, such as a sudden high fever with enlarged lymph nodes, along with the death of rodents during the initial stage of the outbreak. Cases of plague confirmed by pathogenic bacteriological diagnosis or other laboratory evidence were rare. Since the 1950s, China has continuously strengthened plague surveillance and control, leading to a rapid decline in the overall number of human cases, especially in the natural plague foci in Yunnan. This contrasts with the slow decline of natural plague foci among marmot populations in China (20). From 1950 to 2020, the number of human plague cases in the natural plague foci in Yunnan and marmot natural plague foci throughout China (20) exceeded the total number of human plague cases observed in the United States from 1900 to 2012 (21).

The natural plague foci in Yunnan Province are representative of the active plague regions of China (19, 22). A natural focus of plague refers to its unique biological community structure of plague, which constitutes a relatively independent area adapted to the geographical landscape formed under the action of certain climatic factors in a specific environment on the basis of *Y. pestis*, main hosts, vectors, and specific geographical landscape, so it is the basic unit of the epidemic process of plague, and the ecosystem that adapts to its cycle (23). The human plague epidemic in Yunnan Province, China, can be traced back to 1772 and is considered the birthplace of the world's third plague pandemic (24). However, it was not until 1952 that serological and pathogenic bacteriological methods were first used to more accurately diagnose plague in Yunnan Province. Since 1956, the animal epidemic in Yunnan Province has fluctuated, and human cases arise on occasion.

Current human plague cases in China are primarily concentrated in the *Rattus flavipectus* plague focus of the Yunnan, Guangdong, and Fujian provinces, *Apodemus chevrieri-Eothenomys miletus* plague foci of the highlands of northwestern Yunnan Province, and marmot natural plague foci. Sporadic human cases have also appeared in the *Meriones unguiculatus* plague foci of the Inner Mongolian Plateau, along with several recent epidemics among animals that pose a threat to human health in this region (25).

The *Apodemus chevrieri-Eothenomys miletus* plague foci of the highlands of northwestern Yunnan Province are distributed throughout the northeastern part of the canyons in western Yunnan and the western part of the mountain valleys in the Jinsha River Basin, which has an altitude of 2,000 to 3,500 m. The natural foci include three landscape zones, the basin residential area and farmland landscape zone, the pine forest in the middle-high mountains and forest cultivated landscape zone, and the alpine coniferous forest landscape zone. The main hosts are the field mice *Eothenomys miletus* and *Apodemus chevrieri*, and the main vector is the flea, *Neopsylla specialis specialis* (Fig. 1). The *Rattus flavipectus* plague foci of the Yunnan, Guangdong, and Fujian Provinces are mainly distributed in the residential and farmland landscape areas of southwest, central, and eastern Yunnan at an elevation of 400 to 2,100 m. This region includes wide valleys in the west and south, spanning longitudinal valleys in the west, and mountainous areas in the south. The main host is the domestic rat, *Rattus flavipectus*, and the primary vector is the Oriental rat flea, *Xenopsylla cheopis* (Fig. 1).

This study assessed the distribution and characteristics of human plague cases and *Y. pestis* isolated from patients, host animals, and vector insects (mainly fleas) in the

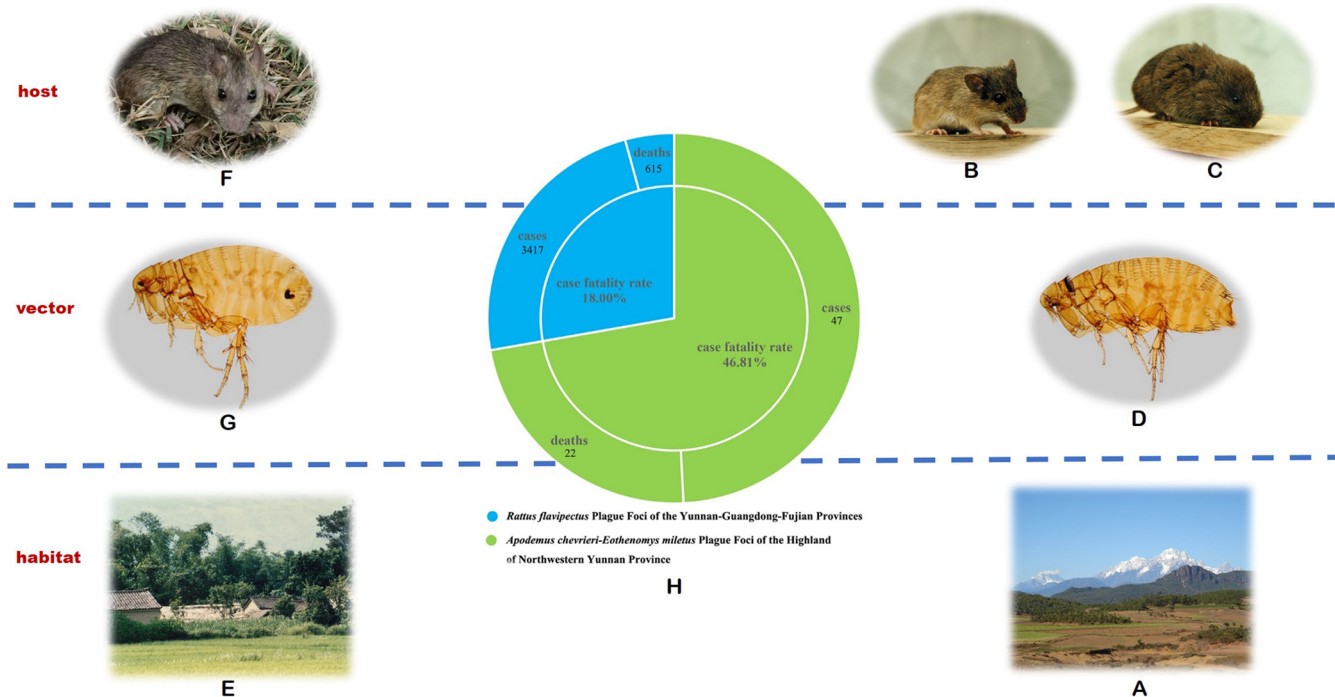

**FIG 1** Habitat, host, vector, and human plague in natural plague foci of Yunnan province. (A) Habitat of *Apodemus chevrieri-Eothenomys miletus* plague foci in the highlands of northwestern Yunnan Province. (B) Host of *Apodemus chevrieri-Eothenomys miletus* plague foci in the highlands of northwestern Yunnan Province, *Apodemus chevrieri*. (C) Host of *Apodemus chevrieri-Eothenomys miletus* plague foci of the highlands of northwestern Yunnan Province, *Eothenomys miletus*. (D) Vector of *Apodemus chevrieri-Eothenomys miletus* plague foci in the highlands of northwestern Yunnan Province, *Neopsylla specialis specialis*. (E) Habitat of *Rattus flavipectus* plague foci of the Yunnan, Guangdong, and Fujian Provinces. (F) Host of *Rattus flavipectus* plague foci of the Yunnan, Guangdong, and Fujian Provinces, *Rattus flavipectus*. (G) Vector of *Rattus flavipectus* plague foci of the Yunnan, Guangdong, and Fujian Provinces, *Xenopsylla cheopis*. (H) Human plague cases, deaths, and case fatality rates in natural plague foci of Yunnan province, 1950 to 2020.

*Rattus flavipectus* plague focus of the Yunnan, Guangdong, and Fujian provinces and the *Apodemus chevrieri-Eothenomys miletus* plague focus of the highlands of northwestern Yunnan Province from 1950 to 2020, and it describes the epidemic situation in these foci after 1950.

## RESULTS

**Distribution characteristics of *Y. pestis* isolates.** From 1952 to 2020, 4,968 *Y. pestis* isolates were obtained from natural plague foci in Yunnan, with an isolation rate of 0.17%. There was an average of 72 isolates obtained each year, with the highest number (*n* = 588) obtained in 1954 (Fig. 2).

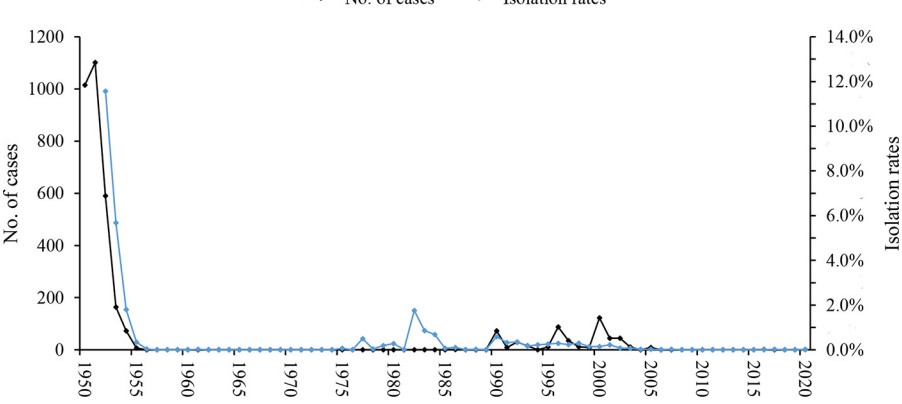

**FIG 2** Human plague (1950 to 2020) cases and *Y. pestis* isolation rates (1952 to 2020) in the natural plague foci of Yunnan Province.

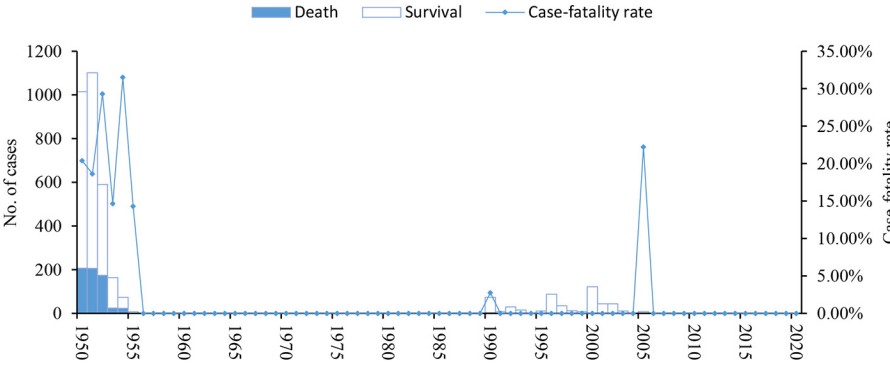

**FIG 3** Frequency of human plague cases and case fatality rate in natural plague foci of Yunnan province, 1950 to 2020.

From 1952 to 2020, 4,747 *Y. pestis* isolates were obtained from the *Rattus flavipectus* plague focus in the Yunnan, Guangdong, and Fujian provinces of Yunnan. There were an average of 68.8 isolates obtained each year, with the highest number ($n = 555$) obtained in 1954. A total of 3,965 *Y. pestis* isolates were obtained from 2,032,191 *Rattus flavipectus* fleas for an isolation rate of 0.2%. The year with the highest isolation rate was 1991 (3.36%). A total of 679 isolates of *Y. pestis* were obtained from 493,538 fleas for an isolation rate of 0.14%. The year with the highest isolation rate was 1952 (0.2%). A total of 103 *Y. pestis* isolates were obtained from 534 patients for an isolation rate of 19.29%. The year with the highest isolation rate was 2000 (40.7%).

A total of 221 *Y. pestis* isolates were obtained from the *Apodemus chevrieri-Eothenomys miletus* plague focus in the highlands of northwestern Yunnan Province. There was an average of 3.2 *Y. pestis* isolates obtained each year, with the highest number obtained in 1983 ($n = 38$). A total of 123 isolates were obtained from 325,795 field mice (*Eothenomys miletus* and *Apodemus chevrieri*) at an isolation rate of 0.04%. The year with the highest isolation rate was 1983 (0.9%). A total of 97 *Y. pestis* isolates were obtained from 123,225 fleas at an isolation rate of 0.08%. The year with the highest isolation rate was 1957 (4.45%). One *Y. pestis* isolate was obtained from five people in 1954.

**Epidemiologic characteristics of the human plague cases in Yunnan.** From 1950 to 2020, 3,464 human cases were diagnosed in the natural plague foci in Yunnan, of whom 637 died, for a total case fatality rate of 18.39%, or 8.97 cases per year. Most plague cases were diagnosed in 1951 (1,101 cases), the highest number of deaths occurred in 1950 (207 deaths), and the highest case fatality rate occurred in 1954 (31.51%) (Fig. 3). The cases spread from the northwest to the southeast of Yunnan during this time period (Fig. 4).

Based on the incidence of human plague in the natural plague foci in Yunnan from 1950 to 2020, the epidemic was divided into four periods, the outbreak period (1950 to 1955), the quiescence period (1956 to 1989), the secondary outbreak period (1990 to 2005), and the sporadic period (2006 to 2020). During the outbreak period, there were 2,950 human plague cases, including 633 deaths, for a case fatality rate of 21.46%, or 491.67 cases per year. During the outbreak period, most plague cases occurred in 1951 (1,101 cases), most deaths occurred in 1950 (207 deaths), and the highest case fatality rate occurred in 1954 (31.51%). During the quiescence period, only one case was found and cured in 1986. During the secondary outbreak period, there were 511 human plague cases, including 4 deaths, for a case fatality rate of 0.78%, or 31.94 cases per year. During the secondary outbreak period, most plague cases occurred in 2000 (122 cases), and the highest case fatality rate occurred in 2005 (22.22%). During the sporadic period, there were two plague cases with no deaths.

From 1950 to 2020, there were 47 human plague cases, including 22 deaths, for a case fatality rate of 46.81% in the *Apodemus chevrieri-Eothenomys miletus* plague focus in the highlands of northwestern Yunnan Province. During the same time, there

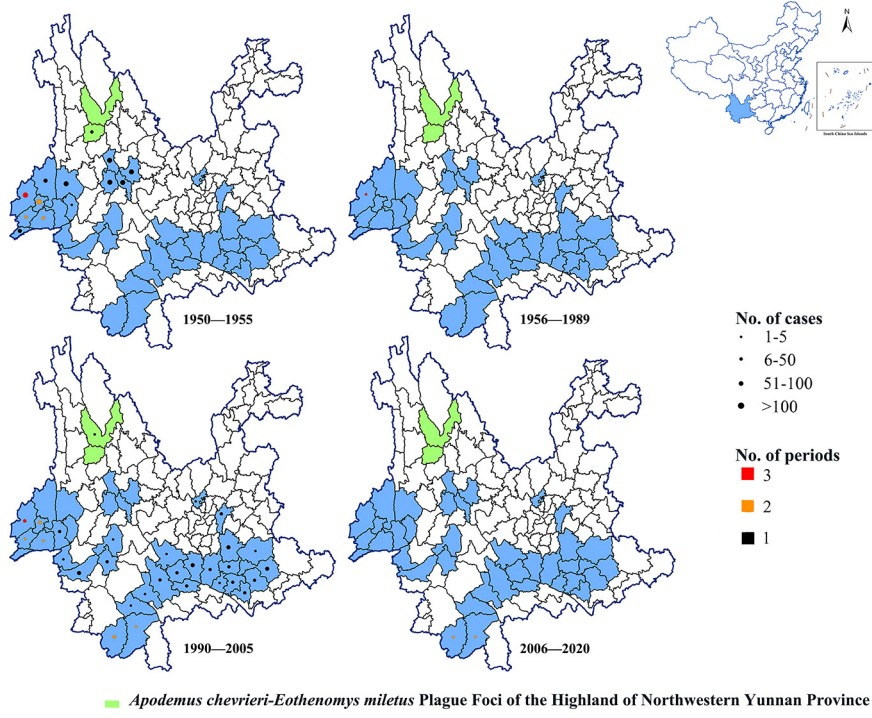

**FIG 4** Spatial distribution of human plague cases during four periods (outbreak period, 1950 to 1955; quiescence period, 1956 to 1989; secondary outbreak period, 1990 to 2005; and sporadic period, 2006 to 2020) in Yunnan Province, China, 1950 to 2020. Dot size indicates the number of human plague cases during which plague occurred in each time frame; dot colors indicate the total number of plague periods at each location during the four periods. Blue shading indicates the *Apodemus chevrieri-Eothenomys miletus* plague foci of the highlands of northwestern Yunnan Province, and green shading indicates the *Rattus flavipectus* plague foci of the Yunnan, Guangdong, and Fujian provinces.

were 3,417 human plague cases, including 615 deaths, for a case fatality rate of 18.00%, in the *Rattus flavipectus* plague focus of the Yunnan, Guangdong, and Fujian provinces (Fig. 1).

**Correlation analysis.** Spearman's rank correlation analysis was conducted between human plague cases and the *Y. pestis* isolation rate in the natural plague foci of Yunnan from 1950 to 2020. The correlation between isolation rate of *Y. pestis* and the number of human plague cases was statistically significant ($r_s = 0.613$, $P < 0.01$), indicating that human plague cases increased as the isolation rate of *Y. pestis* increased (Fig. 2). From 1950 to 2020, the number of human plague cases was statistically related to the isolation rate of *Y. pestis* from domestic rats ($r_s = 0.732$, $P < 0.01$) and the isolation rate of *Y. pestis* from fleas ($r_s = 0.617$, $P < 0.01$) in the *Rattus flavipectus* plague focus in the Yunnan, Guangdong, and Fujian provinces; the number of human plague cases increased as the *Y. pestis* isolation rate increased.

## DISCUSSION

The Yunnan Province of China, as the origin of the world's third plague pandemic (26–28), has attracted widespread attention from researchers. Before 1950, three plague epidemics were recorded in Yunnan, including (i) 1772 to 1855, which was associated with about 253,000 deaths, (ii) 1856 to 1937, which was linked to about 733,500 deaths, and (iii) 1938 to 1949, when 10,450 people became infected, of whom 4,804 people died, for a case fatality rate of 46.5% (29). Although these case numbers are likely inaccurate, these outbreaks were closely associated with a large number of rodent deaths in residential areas. While it is likely that there were some misdiagnoses, however, the accuracy of diagnosis improved over time. In the first two epidemics, for

example, only the number of deaths was recorded, while in the third epidemic, the number of incident cases was also recorded. It is clear that Yunnan Province has been seriously affected by plague and has sustained many casualties since ancient times. It is also likely that plague spreads easily in this region. Thus, it is of great significance to study the epidemic characteristics associated with this area to effectively prevent and control plague outbreaks.

The natural plague focus is a biological community structure that evolves based on the interaction between *Y. pestis*, hosts, vectors, and a specific natural landscape, operating over a relatively independent adaptive geographical landscape area of the plague biological community, influenced by certain climatic factors and geographical environmental conditions, and it is the basic unit of the epidemic process of plague infection and has a certain overall comprehensive effect in a specific biogeographic community (4, 22, 30). The *Apodemus chevrieri-Eothenomys miletus* plague focus of the highlands of northwestern Yunnan Province, discovered in 1974, was centered in Jianchuan County and involved the surrounding counties of Eryuan, Yangbi, Yunlong, Lanping, and Yulong. This is the smallest natural plague focus in China, with nonhibernating small rodents serving as the main host (31). The *Rattus flavipectus* plague focus of the Yunnan, Guangdong, and Fujian provinces is widely distributed in China. This focus used to be prevalent in 276 counties (cities) in 9 provinces south of the Yangtze River. However, in the past 70 years, this natural focus has only been prevalent in Yunnan Province (32). According to mortality data of humans, studies suggest that the *Y. pestis* isolates obtained from *Eothenomys miletus* and *Apodemus chevrieri* are significantly more virulent than those found in *Rattus flavipectus* (33). In addition, another piece of strong evidence for the virulence of the isolates from the above-mentioned two foci was that 0.5 mL of *Y. pestis* suspension isolated from these two foci was, respectively, injected subcutaneously into the left groin of guinea pigs, which results showed that the median lethal dose ($LD_{50}$) of guinea pigs to *Y. pestis* isolated from the *Eothenomys miletus* was 6.7, and the average survival time was 7.5 days, while the $LD_{50}$ of guinea pigs against *Y. pestis* isolated from *Rattus flavipectus* was 10.96, and the average survival time was 11.94 days (34, 35). Genome analysis has confirmed that *Y. pestis* isolates isolated from domestic rats are the most recent strains associated with the natural plague foci in Yunnan (24). This region has an average annual temperature of 19 to 20°C, allowing plague to occur among animal populations throughout the year. Yunnan is rich in animal and plant species, especially in the Hengduan Mountains, where there are clear horizontal and vertical changes in climate, soil, vegetation, topography, and landforms, forming a polymorphic landscape that promotes organism diversity and the development of natural plague foci. This explains why the incidence of plague hosts, vectors, *Y. pestis* types, and other organisms in this region is significantly higher than that found in other areas, allowing natural plague foci to form a natural barrier to remain protected (36). Many kinds of plague-infected animals are found in the foci, most of which are *Rattus flavipectus*, where, rarely, pneumonic plague cases were reported; the virulence of the plague strain in this foci is relatively weak (37). The natural plague foci in Yunnan, a relatively active area for animal and human plague in China, have a similar ecological structure and range of host vectors as those in the natural plague foci of Madagascar (38, 39).

Plague epidemics within the natural plague foci share the characteristics of localization, dispersion, and concealment. This means that foci may be composed of numerous small foci, each of which forms its own epidemic and quiescent patterns. If there are no external interferences or environmental changes, the plague may complete its circulation in the small focus, which is consistent with the endogenous preservation mechanism of *Y. pestis*. This pathogen circulates between its host and vector with limited range and uneven distribution, rarely causing disease or being detected by regular monitoring (19, 40–42). Some studies believe that plague can continue circulating at a low level in a focus because infected species are either relatively resistant or highly susceptible and because the population has a high replacement rate (43). During the third

pandemic, the plague achieved long-distance transmission through sea transportation, forming many new natural plague foci. For example, the ecological environment in North America allowed for the establishment of several new and stable natural foci where there were enough native rodents and the environment was suitable for *Y. pestis* (4). However, when an epidemic broke out in Australia, it lacked a suitable ecological environment or hosts and subsided after the number of nonnative rodents decreased (44). This indicates that an appropriate ecological environment and host types are prerequisites for the persistence of *Y. pestis*. Studies show that plague ecology consists of two main stages, a rapid expansion stage accompanied by an increase in the number of bacterial groups and a slow and persistent bacterial species dispersal stage, both of which are required for the long-term preservation of *Y. pestis*. To maintain quiescent *Y. pestis*, both the host animals and vector fleas play a critical role (41, 45). Since the small rodent hosts of the natural plague foci in Yunnan have both a high replacement rate and high susceptibility, Yunnan is an excellent natural focus for the low-level circulation of *Y. pestis*.

In addition to the highly susceptible large rodent marmot, hosts of the *Marmota himalayana* plague foci in the Qinghai-Tibet Plateau include mammals such as canines and cloven-hoofed animals such as the Tibetan antelope (46), which are significantly larger and live longer than the hosts in the natural plague foci of Yunnan, which suggests that the hosts might have a longer life span than the small rodents in Yunnan. During the past 70 years in China, the average fatality rate of human plague cases infected by *M. himalayana* in the *M. himalayana* plague foci of the Qinghai-Tibet Plateau was high (68.88%), and the plague was so virulent because bubonic plague becomes pneumonic plague after a brief time period (47), and the main mode of transmission was through wounds incurred while skinning infected marmots (48). The primary and secondary pneumonic plagues, the types of plague associated with the highest mortality, accounted for a high proportion of cases in the *M. himalayana* plague foci in the Qinghai-Tibet Plateau (49). The case fatality rate caused by pneumonic plague (75.10%) is significantly higher than it is for other types of plague (54.88%) (29). Pneumonic plague becomes a dangerous respiratory infection when it is secondary to bubonic or other types of plague, and its natural focus characteristics gradually disappear (37, 50, 51). This is verified by the fluctuating nature of the *M. himalayana* plague foci (20).

The pathogen also influences the natural foci. Genomics comparative analysis found that *Y. pestis* strains from the *Apodemus chevrieri-Eothenomys miletus* plague focus of the highlands of northwestern Yunnan Province, which have been epidemic in these foci for many years, are homologous to strains in the *M. himalayana* plague foci of the Qinghai-Tibet Plateau. At the same time, there were significant differences between the *Y. pestis* stains from the *Apodemus chevrieri-Eothenomys miletus* plague focus of the highlands of northwestern Yunnan Province and the *Rattus flavipectus* plague focus of the Yunnan, Guangdong, and Fujian provinces in Yunnan, which may have been formed during the transmission and evolution of *Y. pestis* (52–55). It takes a long time for *Y. pestis* to develop several mutations (56). In evolutionary time, *Y. pestis* strains from Lijiang City appeared earlier than those from Jianchuan County, and *Y. pestis* strains from Jianchuan County appeared earlier than those from the *Rattus flavipectus* plague focus of the Yunnan, Guangdong, and Fujian provinces. The *Y. pestis* strains associated with the *Rattus flavipectus* plague focus of the Yunnan, Guangdong, and Fujian provinces are the most recent and youngest strains in Yunnan (24). Studies indicate that *Y. pestis* strains from Lijiang City and Jianchuan County have unique genetic characteristics that are similar to *Y. pestis* strains from the *M. himalayana* plague foci of the Qinghai-Tibet Plateau, each of which has formed an independent circular process because it has remained epidemic in the local natural plague foci for a long period (54, 57–59). The biological characteristics, differences, and adaptive changes in the environmental gradients of the entire plague biogeographic community, as well as the genetic differences, evolutionary relationships, epidemiological

characteristics, and regional distributions of these pathogens, make it clear that the natural plague foci in Yunnan originated from the *M. himalayana* plague foci of the Qinghai-Tibet Plateau and gradually formed along the Sanjiang River Valley. The natural plague foci evolved from the *M. himalayana* plague foci in the Qinghai-Tibet Plateau, to the natural focus of the Yulong plague in Lijiang, to the *Apodemus chevrieri-Eothenomys miletus* plague focus of the highlands of northwestern Yunnan Province, and finally to the *Rattus flavipectus* plague focus of the Yunnan, Guangdong, and Fujian provinces (40, 60).

The stability of plague natural foci is closely related to the pathogen and host but is also influenced by local environmental factors. Natural disasters or human-induced environmental changes can cause local ecological changes in the natural plague foci that lead to changes in the density of host animals and vector insects and even the phenotype and gene mutations of *Y. pestis*. This can ultimately destroy the balance of the plague natural focus ecosystem and prompt outbreaks of animal or human plague or the activation of plague natural foci from its quiescent state (61). It is also reported that the intensity of the plague epidemic decreased and then increased with a rise in humidity, intensifying when the humidity was extremely high (flood disaster) or extremely low (drought). Indeed, the epidemic intensity of plague in a particular year is directly impacted by the intensity of humidity or drought that occurs the previous year (62). This can result from a rise in precipitation or dry weather, both of which interfere with the availability of food and strongly affect the activity of rodent populations (22, 62). The prevalence of plague in natural plague foci is associated with very complex and comprehensive factors and is difficult to explain with a single indicator. For example, in the *M. himalayana* plague foci in the Altun Mountains of the Qinghai-Tibet Plateau in Gansu, China, continuous drought increases the epidemic intensity of plague among animals. However, in years with more rain, the epidemic intensity decreases because in this harsh plateau environment, the main host, *M. himalayana*, lacks food, leading to poorer health and greater susceptibility to infection (63). Ecological changes also lead to differences in the number and density of hosts in the foci, affecting the pathogen reservoir (19).

Based on time-segmented data relating to *Y. pestis* isolation over the past 70 years, cases in Yunnan Province peaked from 1950 to 1951, when isolation technology and animal surveillance were poor. During this time, plague cases were primarily diagnosed using clinical and epidemiological investigations, and no *Y. pestis* isolates could be used for laboratory confirmation. With the establishment of laboratory testing technology and personnel training, *Y. pestis* culturing began in 1952. (While Robert Pollitzer, Tang Feifan, and others had dissected dead rats and patients and performed bacteriological culture in 1940, the first recorded bacteriological confirmation of the Yunnan plague, systematic bacteriological surveillance data were lacking before 1952). Since then, the number of *Y. pestis* isolates has continued to increase and closely correlates with the number of plague cases (Fig. 2). Plague surveillance in animals (64) is still significant for the prevention and control of local plague outbreaks in Yunnan.

Based on the characteristics of 71 human plague cases in Yunnan Province from 1950 to 2020, the epidemic can be divided into four stages. The first, occurring from 1950 to 1955, was the outbreak stage. During this period, the regions of Yunnan with natural plague foci had poor economic conditions and very weak medical resources, people lacked awareness of the plague, and treatment access was limited. A considerable number of patients developed septicemic or pneumonic plague. The pneumonic plague caused a vicious circle of extensive human-to-human transmission from patients, resulting in a large outbreak with a high fatality rate (21.46%).

The second stage, occurring from 1956 to 1989, was the quiescent period. During this stage, some studies indicate that under certain environmental conditions, the plague will gradually enter a longer period of quiescence after the outbreak ends (65). At this stage, both human and animal plague are in a relatively quiet stage. From 1975 to 1982, the detection rate of *Y. pestis* in Yunnan began to increase, reaching a peak of 1.76%, and from 1982 to 1989, the detection rate began to decline. In 1986, a human

plague case occurred in the natural plague foci in Yunnan, indicating that even without the occurrence of human plague for years, animal plague can continue to spread. This provides evidence for low-level circulation of plague in Yunnan. Previous studies also believe that the interaction between *Y. pestis* and the environment, weather, hosts, insect vectors, and other factors in the natural foci constitute a relative balance so that when the plague appears and disappears in the natural foci, rest and activity alternate (20). The spread of *Y. pestis* is very limited, forming multiple tiny foci with discrete distribution. During this stage, the infection is in a long dormant period and can even be difficult to detect. In 2000, the construction of a hydropower station at the junction of four provinces in China led to an epidemic of plague among animals, which also infected 210 people. Importantly, a plague-like case occurring here nearly 100 years previously resulted in more than 10,000 deaths (19). This alternation of rest and activity favors the continued circulation of the plague pathogen.

The third stage, occurring from 1990 to 2005, was the second outbreak stage, which broke the internal balance of the natural plague foci because of changes in the natural environment and the intervention of human factors. This led to an increase in the density of rats and a continuous increase in the host and vector populations, allowing the spread between various tiny foci. A resurgence epidemic is the result of the coexistence between plague resurgence and spread.

The fourth stage, occurring from 2006 to 2020, was the sporadic stage. This likely occurred because human activities affected many scattered natural plague foci during the quiescent period, causing dramatic changes to the ecological environment in a short period of time. Animal migration also led to changes in the density and distribution of host populations. Scattered natural foci of *Y. pestis* aggregate and eventually merge with other foci, and previously dispersed *Y. pestis* foci expand, increasing the risk of transmission between rodents and humans. Natural plague foci return to the quiescent stage as the host adapts to the new habitat and returns to a stable state. This highly supports the theory of an endogenous bacterial protection mechanism that allows *Y. pestis* to persist at a low level in the natural plague foci during the quiescent stage, which is difficult to detect (19, 66). While human plague did not occur on a large scale as a result of human intervention, plague isolates were still detected in the host animals, and real-time surveillance of the natural plague foci is required.

The outbreak stage from 1950 to 1955 was primarily concentrated in the western region around the *Apodemus chevrieri-Eothenomys miletus* plague focus of the highlands of northwestern Yunnan Province. During this stage, there were 2,950 prevalent cases in which 633 individuals died, with a case fatality rate of 21.46%. During the second outbreak stage from 1990 to 2005, there was a transition from the western region around the *Apodemus chevrieri-Eothenomys miletus* plague focus of the highlands of northwestern Yunnan Province to the southern region around the *Rattus flavipectus* plague focus of the Yunnan, Guangdong, -and Fujian provinces in Yunnan. There were 511 prevalent cases and 4 deaths, for a case fatality rate of 0.78%. Furthermore, some studies have suggested that wild rodents are riskier than urban commensal rodents to humans (67), and the mortality data from these two foci illustrate this point, so we believe that the *Y. pestis* isolates from *Eothenomys miletus* and *Apodemus chevrieri* in the *Apodemus chevrieri-Eothenomys miletus* plague focus of the highlands of northwestern Yunnan Province were more virulent than those from *Rattus flavipectus* in the *Rattus flavipectus* plague focus of the Yunnan, Guangdong, and Fujian provinces. This finding emphasizes the importance of confirming plague patients by pathogenic bacteriological diagnosis or other laboratory results. At the same time, the government invested more into strengthening surveillance in two ways. First, animal and human plague monitoring and reporting systems were consolidated and improved, while several actions were taken to improve community awareness about prevention and control efforts within the foci. When sick and dead mice and suspected cases of plague were found, they were actively reported. Second, several disciplines and technologies

were used to explore the distribution of plague within the foci and to analyze and screen important factors affecting the epidemic (40).

Rodent and flea eradication activities were carried out to effectively control the occurrence of plague among humans and animals. Control measures for the *Rattus flavipectus* plague focus of the Yunnan, Guangdong, and Fujian provinces included the formation of an emergency response that focused on controlling the source of infection, cutting off the route of transmission, and protecting healthy people. Control measures for the *Apodemus chevrieri-Eothenomys miletus* plague focus of the highlands of Northwestern Yunnan province were taken to reduce the density of rodents and fleas in plague areas of endemicity and surrounding residential areas to protect the population and lower the risk of spread (68). As a result of expanded knowledge about plague prevention and control efforts, increased publicity of relevant laws and regulations by local government and relevant departments (69), continuous improvement of self-protection efforts by local residents in the foci, and timely diagnosis and clinical treatment of *Y. pestis* infections (70), the number of patients declined rapidly, with only occasional outbreaks leading to localized plague epidemics.

From 1950 to 2020, human plague cases in the natural foci of Yunnan Province, China, were mainly distributed throughout the *Rattus flavipectus* plague focus of the Yunnan, Guangdong, and Fujian provinces and, to a far lesser extent, the *Apodemus chevrieri-Eothenomys miletus* plague focus of the highlands of northwestern Yunnan Province. The *Apodemus chevrieri-Eothenomys miletus* plague focus in the highlands of northwestern Yunnan Province had a higher fatality rate than the *Rattus flavipectus* plague focus of the Yunnan, Guangdong, and Fujian provinces (46.81% and 18.00%, respectively). These natural foci are alternately active and quiescent, further illustrating the endogenous preservation mechanism of plague (71, 72). The current study analyzed human cases and animal epidemics of plague in the natural foci of Yunnan from 1950 to 2020 from the perspective of time, space, and population. In 1952, a special plague investigation agency was established to strengthen the surveillance and control of plague in this region, resulting in a decline in human cases. Cases associated with the Yunnan plague foci declined more quickly than those linked to the *M. himalayana* plague foci of the Qinghai-Tibet Plateau (20). Considering the endogenous preservation mechanism of natural plague foci, including their alternating quiescence and activity, prevention and control measures should not be relaxed, even during this downward trend.

## MATERIALS AND METHODS

**Data source analysis.** This study included human plague cases from 1950 to 2020 and *Y. pestis* isolates from 1952 to 2020 associated with the *Rattus flavipectus* plague focus of the Yunnan, Guangdong, and Fujian provinces and the *Apodemus chevrieri-Eothenomys miletus* plague focus of the highlands of northwestern Yunnan Province in Yunnan Province, China. The data were primarily obtained from the history of spread of plague in China (volumes 1 and 2) (29) compiled by the Institute of Epidemiology and Microbiology at the Chinese Academy of Medical Sciences (now known as the Institute for Infectious Disease Control and Prevention, Chinese Center for Disease Control and Prevention) in 1981 and historical plague surveillance data from 1950 to 2020. The isolates were obtained from patients, host animals, and vector insects (mainly fleas) from the natural plague foci in Yunnan, China. The data were collected and used to describe the distribution range and characteristics of human plague cases and *Y. pestis* isolates in the natural plague foci in Yunnan and to predict the epidemic trend of plague in this region. Statistical methods included epidemiological distribution, correlation analysis, and geographic informatics.

**Diagnosis of plague cases.** Human cases in this study were diagnosed based on clinical symptoms, epidemiological investigations, and etiology and serology methods. Before 1952, plague cases in the natural foci in Yunnan, China, were primarily diagnosed using clinical symptoms and epidemiological investigations. Since 1952, etiology and serology methods have been used to diagnose plague cases in the natural foci of Yunnan.

**Data availability.** All data generated or analyzed during this study are presented within the manuscript and figures.

## ACKNOWLEDGMENTS

We thank Charlesworth Author Services (paper no. 97063) for their critical editing and helpful comments regarding our manuscript.

This work was supported by the National Science and Technology Major Project (2018ZX10713-003-002, 2018ZX10713-001-002). The funding sources for this study had no role in the design of the study and collection, analysis, interpretation of data, and in writing the manuscript.

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
