## [Reviewer comments · Microbiology Spectrum]

Microbiology Spectrum

Epidemiological characteristics of human and animal plague in Yunnan Province, China, 1950-2020

Haonan Han, Yun Liang, Zhizhong Song, Zhaokai He, Ran Duan, Yuhuang Chen, Zihou Gao, shuai qin, Junrong Liang, Deming Tang, Dongyue Lv, Peng Zhang, Dan Zhang, Huaiqi Jing, and Xin Wang

Corresponding Author(s): Xin Wang, National Institute for Communicable Disease Control and Prevention, Chinese Center for Disease Control and Prevention

Review Timeline:

Submission Date:	May 12, 2022
Editorial Decision:	June 10, 2022
Revision Received:	July 11, 2022
Editorial Decision:	August 9, 2022
Revision Received:	September 7, 2022
Accepted:	September 15, 2022

Editor: Sadjia Bekal

Reviewer(s): Disclosure of reviewer identity is with reference to reviewer comments included in decision letter(s). The following individuals involved in review of your submission have agreed to reveal their identity: Andrey P. Anisimov (Reviewer #1)

Transaction Report:

DOI: <https://doi.org/10.1128/spectrum.01662-22>

June 10, 2022

Prof. Xin Wang
National Institute for Communicable Disease Control and Prevention, Chinese Center for Disease Control and Prevention
State Key Laboratory of Infectious Disease Prevention and Control
Changbai Road 155
Beijing 102206
China

Re: Spectrum01662-22 (Epidemiological characteristics of human and animal plague in Yunnan Province, China, 1950-2020)

Dear Prof. Xin Wang:

Link Not Available

Sincerely,

Sadjia Bekal

Journals Department
Reviewer comments:

Reviewer #1 (Comments for the Author):

When you write about the virulence of strains, do not forget to indicate in relation to which animal species this indicator is given. Without specifying the species and even the breed of the biomodel, this indicator is meaningless.

Reviewer #2 (Comments for the Author):

1. The characteristics of the natural plague foci in China (p. 3.1. of the Results section) have been described previously (for

example, Wang YS, et al, 2007). This information would be more appropriate in the Introduction section.

2. There are repetitions in the manuscript: L 67-69, 85-88, 244-246; L 96, 160

3. L 64 "multiple plague pandemics occurred in China" - epidemics

4. L 91 "meteorological and environmental factors" - climatic

5. L 76, 96, 160, 440 The term "etiological" is used incorrectly in the meaning of the term "bacteriological".

6. L 213-214 "...indicating that human plague cases increased as the isolation rate of *Y. pestis* increased (Figure 4)." - Maybe it is from Figure 2

7. Figure 4 - the histogram at the top of the figure duplicates the data in Figure 2. In my opinion, the figure will be more informative if the histogram is deleted, and the size of the plague cases distribution maps is increased. In the current view, the size and color of points on the map are not visible.

Staff Comments:

Preparing Revision Guidelines

Please return the manuscript within 60 days; if you cannot complete the modification within this time period, please contact me. If you do not wish to modify the manuscript and prefer to submit it to another journal, please notify me of your decision immediately so that the manuscript may be formally withdrawn from consideration by Microbiology Spectrum.

Reviewer #1 (Comments for the Author):

Question(Q1):

When you write about the virulence of strains, do not forget to indicate in relation to which animal species this indicator is given. Without specifying the species and even the breed of the biomodel, this indicator is meaningless.

Answers(A1):

Thanks for reviewer's comments. We have marked the source animals for the strains related virulence mentioned in the text according to your suggestion. The original text is revised as follows:

L 218-220: “Studies indicate that the virulence of *Y. pestis* isolates obtained from *Eothenomys miletus* and *Apodemus chevrieri* are significantly more virulent than those found in *Rattus flavipectus*.”

L 230-232: “Many kinds of plague-infected animals are found in the foci, most of which are *Rattus flavipectus*, and the virulence of the plague strains in this region is relatively weak.”

L 266-269: “During the past 70 years in China, the average fatality rate of human plague cases infected by *M. himalayana* in the *M. himalayana* plague foci of the Qinghai-Tibet Plateau was high (68.88%), where the plague was so virulent because bubonic plague becomes pneumonic plague after a brief time period.”

L 401-405: The *Y. pestis* isolates from *Eothenomys miletus* and *Apodemus*

chevrieri in the *Apodemus chevrieri-Eothenomys miletus* plague focus of the highland of Northwestern Yunnan Province were more virulent than those from *Rattus flavipectus* in the *Rattus flavipectus* plague focus of the Yunnan-Guangdong-Fujian provinces.

Reviewer #2 (Comments for the Author):

Question(Q1):

1. The characteristics of the natural plague foci in China (p. 3.1. of the Results section) have been described previously (for example, Wang YS, et al, 2007). This information would be more appropriate in the Introduction section.

Answers(A1):

We completely agree with the reviewer's comments. We have placed the characteristics of the natural plague foci in China (p. 3.1. of the Results section) after the third paragraph of the Introduction section as you suggested in L104-118.

Question(Q2):

2. There are repetitions in the manuscript: L 67-69, 85-88, 244-246; L 96, 160

Answers(A2):

Thanks to the editor for the kind reminder. We have condensed the above

sections as you suggested. The content you mentioned above has been edited and changed by us to make it more accurate. We have simplified the above parts as you suggested. The original text is revised as follows:

L 66-70: Of these, two natural plague foci have been identified and classified in Yunnan province, that is, the *Rattus flavipectus* plague focus of the Yunnan-Guangdong-Fujian provinces and the *Apodemus chevrieri-Eothenomys miletus* plague focus of the highland of Northwestern Yunnan Province, and these two foci cover approximately 79,216 km² (originally in L 67-69).

In addition, to avoid repetitions, we have deleted 3 sentences as you remind us. Details are as follows:

2.1 We deleted the sentence originally in L 85-88, “There are two kinds of natural foci of plague in Yunnan, *Rattus flavipectus* plague focus of the Yunnan-Guangdong-Fujian provinces and *Apodemus chevrieri-Eothenomys miletus* plague focus of the highland of Northwestern Yunnan Province.”

2.2 We deleted the sentence originally in L 244-246, “To date, two natural plague foci have been identified and classified in Yunnan province, that is, *Rattus flavipectus* plague focus of the Yunnan-Guangdong-Fujian provinces and *Apodemus chevrieri-Eothenomys miletus* plague focus in the highland of Northwestern Yunnan Province.”

2.3 We deleted the sentence originally in L 160, “It was not until 1952

that the etiological culture of *Y. pestis* began in Yunnan.”

Question(Q3):

3. L 64 "multiple plague pandemics occurred in China" - epidemics

Answers(A3):

Thanks for reviewer's comments. We have replaced "multiple plague pandemics" with "epidemics" as you suggested in L 63. The original text is revised as follows: “In the first half of the 20th century, plague epidemics occurred in China, and the death toll reached nearly one million.”

Question(Q4):

4. L 91 "meteorological and environmental factors" – climatic

Answers(A4):

Thanks for reviewer's recommendation. We have replaced "meteorological and environmental factors" with "climatic" as you suggested which in L 88 in "Marked Up Manuscript - For Review Only". The original text is revised as follows: “A natural focus of plague refers to its unique biological community structure of plague, which constitutes a relatively independent area adapted to the geographical landscape formed under the action of certain climatic factors, in a specific environment on the basis of *Y. pestis*, main hosts, vectors and specific

geographical landscape, so it is the basic unit of the epidemic process of plague and the ecosystem that adapts to its cycle.”

Question(Q5):

5. L 76, 96, 160, 440 The term "etiological" is used incorrectly in the meaning of the term "bacteriological".

Answers(A5):

Thanks for reviewer's comments. We have replaced the term "etiological" or “pathogenic” with the term "pathogenic bacteriological" as you suggested. The original text is revised as follows:

L 77: “Cases of plague confirmed by pathogenic bacteriological diagnosis or other laboratory evidence were rare (originally in the L76)”.

L 94: “However, it was not until 1952 that serological and pathogenic bacteriological methods were first used to more accurately diagnose plague in Yunnan Province (originally in the L96).”

L406: “This finding emphasizes the importance of confirming plague patients by pathogenic bacteriological diagnosis or other laboratory results (originally in the L440).”

In addition, to avoid repetitions, we have deleted the sentence, “It was not until 1952 that the etiologial culture of *Y. pestis* began in Yunnan.”, originally in the L160.

Question(Q6):

6. L 213-214 "...indicating that human plague cases increased as the isolation rate of *Y. pestis* increased (Figure 4)." - Maybe it is from Figure 2.

Answers(A6):

Thanks for the editor's reminding. We are sorry for the mistake. According to your suggestion, "Figure 4" in the content of the original text has been changed into "Figure 2" in L 181.

Question(Q7):

7. Figure 4 - the histogram at the top of the figure duplicates the data in Figure 2. In my opinion, the figure will be more informative if the histogram is deleted, and the size of the plague cases distribution maps is increased. In the current view, the size and color of points on the map are not visible.

Answers(A7):

Thanks for reviewer's suggestion. According to your suggestion, we removed the histogram and changed the Figure 4. After removing the histogram, the figure become more focused and the size of the plague cases distribution maps is increased as you suggested. The modified figure (thumbnail version) is as follows:

Figure 4. Spatial distribution of human plague cases during four periods (Outbreak Period, 1950–1955, Quiescence Period, 1956–1989, Secondary Outbreak Period, 1990–2005, and Sporadic Period, 2006–2020) in Yunnan Province, China, 1950-2020. Dot size indicates the number of human plague cases during which plague occurred in each timeframe; dot colors indicate the total number of plague periods at each location during the four periods. Blue shading indicates the *Apodemus chevrieri-Eothenomys miletus* plague foci of the highland of Northwestern Yunnan

Province and green shading indicates the *Rattus flavipectus* plague foci of the Yunnan-Guangdong-Fujian Provinces.

If there are still any problems, we are willing to listen to further suggestions from reviewers and editors.

August 9, 2022

Prof. Xin Wang

National Institute for Communicable Disease Control and Prevention, Chinese Center for Disease Control and Prevention
State Key Laboratory of Infectious Disease Prevention and Control
Changbai Road 155
Beijing 102206
China

Re: Spectrum01662-22R1 (Epidemiological characteristics of human and animal plague in Yunnan Province, China, 1950-2020)

Dear Prof. Xin Wang:

Link Not Available

Sincerely,

Sadjia Bekal

Journals Department
Reviewer comments:

Reviewer #1 (Comments for the Author):

Answers(A1):

Thanks for reviewer's comments. We have marked the source animals for the strains related virulence mentioned in the text according to your suggestion. The original text is revised as follows:

L 218-220: "Studies indicate that the virulence of *Y. pestis* isolates obtained from *Eothenomys miletus* and *Apodemus chevrieri* are significantly more virulent than those found in *Rattus flavipectus*."

Perhaps I didn't phrase my question clearly enough. I will try again. It is clear that strains isolated from *Eothenomys miletus* and

Apodemus chevrieri are more virulent. However, it is necessary to additionally indicate for which animal species they were more or less virulent, for Eothenomys miletus, Apodemus chevrieri, Rattus flavipectus, laboratory mouse, guinea pigs or man.

Line 232 ... the virulence of the plague strains in this region is relatively weak (35).
For which animal species is virulence low?

L 401-405: The Y. pestis isolates from Eothenomys miletus and Apodemus chevrieri in the Apodemus chevrieri-Eothenomys miletus plague focus of the highland of Northwestern Yunnan Province were more virulent than those from Rattus flavipectus in the Rattus flavipectus plague focus of the Yunnan-Guangdong-Fujian provinces.
For which animal species is virulence low?

Reviewer #2 (Comments for the Author):

I thank the authors for taking into account all my comments.

Staff Comments:

Preparing Revision Guidelines

Please return the manuscript within 60 days; if you cannot complete the modification within this time period, please contact me. If you do not wish to modify the manuscript and prefer to submit it to another journal, please notify me of your decision immediately so that the manuscript may be formally withdrawn from consideration by Microbiology Spectrum.

These issues still require clarification.

Answers(A1):

Thanks for reviewer's comments. We have marked the source animals for the strains related virulence mentioned in the text according to your suggestion. The original text is revised as follows:
L 218-220: "Studies indicate that the virulence of *Y. pestis* isolates obtained from *Eothenomys miletus* and *Apodemus chevrieri* are significantly more virulent than those found in *Rattus flavipectus*."

Perhaps I didn't phrase my question clearly enough. I will try again. It is clear that strains isolated from *Eothenomys miletus* and *Apodemus chevrieri* are more virulent. However, it is necessary to additionally indicate for which animal species they were more or less virulent, for *Eothenomys miletus*, *Apodemus chevrieri*, *Rattus flavipectus*, laboratory mouse, guinea pigs or man.

Line 232 ... the virulence of the plague strains in this region is relatively weak (35).
For which animal species is virulence low?

L 401-405: The *Y. pestis* isolates from *Eothenomys miletus* and *Apodemus chevrieri* in the *Apodemus chevrieri*-*Eothenomys miletus* plague focus of the highland of Northwestern Yunnan Province were more virulent than those from *Rattus flavipectus* in the *Rattus flavipectus* plague focus of the Yunnan-Guangdong-Fujian provinces.
For which animal species is virulence low?

Dear Editors:

I am sending a new revised manuscript entitled “Epidemiological characteristics of human and animal plague in Yunnan Province, China, 1950-2020” (Spectrum01662-22) by Haonan Han et al, which I should like to submit for possible publication in the Microbiology Spectrum. The revised manuscript indicates the changes from the original submission by highlighting in yellow as file type "Marked Up Manuscript - For Review Only". We believe the paper may be of particular interest to the readers of your journal to summarize and analyze the epidemic pattern of human plague and the distribution characteristics of *Y. pestis* in the natural plague foci in Yunnan, providing a scientific basis for further development and adjustment of plague prevention and control strategies.

Thanks for the reviewers' nice comments and suggestions, we have learned a lot from them, and thanks for editors. Thank you very much for your attention and consideration.

Yours sincerely,

Xin Wang

Reviewer comments:

Reviewer #1 (Comments for the Author):

Answers(A1):

Thanks for reviewer's comments. We have marked the source animals for the strains related virulence mentioned in the text according to your suggestion. The original text is revised as follows:

L 218-220: "Studies indicate that the virulence of *Y. pestis* isolates obtained from *Eothenomys miletus* and *Apodemus chevrieri* are significantly more virulent than those found in *Rattus flavipectus*."

Perhaps I didn't phrase my question clearly enough. I will try again. It is clear that strains isolated from *Eothenomys miletus* and *Apodemus chevrieri* are more virulent. However, it is necessary to additionally indicate for which animal species they were more or less virulent, for *Eothenomys miletus*, *Apodemus chevrieri*, *Rattus flavipectus*, laboratory mouse, guinea pigs or man.

Line 232 ... the virulence of the plague strains in this region is relatively weak (35).

For which animal species is virulence low?

L 401-405: The *Y. pestis* isolates from *Eothenomys miletus* and *Apodemus chevrieri* in the *Apodemus chevrieri*-*Eothenomys miletus* plague focus of

the highland of Northwestern Yunnan Province were more virulent than those from *Rattus flavipectus* in the *Rattus flavipectus* plague focus of the Yunnan-Guangdong-Fujian provinces.

For which animal species is virulence low?

Answers:

Thanks for your nice comment and patient explanation again. In response to your suggestion, I have reconsidered my words and added relevant information.

L 218-228: According to mortality data of human, studies suggest that the virulence of *Y. pestis* isolates obtained from *Eothenomys miletus* and *Apodemus chevrieri* are significantly more virulent than those found in *Rattus flavipectus*(33). In addition, another strong evidence for the virulence of the isolates from abovementioned two foci was that 0.5ml of *Y. pestis* suspension isolated from these two foci was respectively injected subcutaneously into the left groin of guinea pigs, which results showed that the median lethal dose (LD₅₀) of guinea pigs to *Y. pestis* isolated from the *Eothenomys miletus* was 6.7, and the average survival time was 7.5 days, while the median lethal dose (LD₅₀) of guinea pigs against *Y. pestis* isolated from *Rattus flavipectus* was 10.96 *Y. pestis*, and the average survival time was 11.94 days (34, 35).

L 239: ... where rarely pneumonic plague cases were reported, they are considered the virulence of the plague strains in this region is relatively weak (37).

L 409-415: Furthermore, some studies have suggested that wild rodents are more risk than urban commensal rodents to humans (67), and the mortality data of human from these two foci illustrate this point, so we believe that the *Y. pestis* isolates from *Eothenomys miletus* and *Apodemus chevrieri* in the *Apodemus chevrieri*-*Eothenomys miletus* plague focus of the highland of Northwestern Yunnan Province were more virulent than those from *Rattus flavipectus* in the *Rattus flavipectus* plague focus of the Yunnan-Guangdong-Fujian provinces.

Reviewer #2 (Comments for the Author):

I thank the authors for taking into account all my comments.

Answer: Thanks for your nice comment and kind help again.

If there are still any problems, we are willing to listen to further suggestions from reviewers and editors.

September 15, 2022

Prof. Xin Wang
National Institute for Communicable Disease Control and Prevention, Chinese Center for Disease Control and Prevention
State Key Laboratory of Infectious Disease Prevention and Control
Changbai Road 155
Beijing 102206
China

Re: Spectrum01662-22R2 (Epidemiological characteristics of human and animal plague in Yunnan Province, China, 1950-2020)

Dear Prof. Xin Wang:

Your manuscript has been accepted, and I am forwarding it to the ASM Journals Department for publication. You will be notified when your proofs are ready to be viewed.

Sincerely,

Sadjia Bekal
Editor, Microbiology Spectrum
